# Liver-specific glucagon dysfunction promotes PP-cell hyperplasia and formation of glucagon and PP double-positive cells

Yuko Nakagawa[1], Takuro Horii[2], Ayako Fukunaka[1], Takashi Sato[1], Zhehao Zhang[1], Masaki Kobayashi[3], Tadahiro Kitamura[3], Yoshitaka Hayashi[4], Takashi Nishimura[5], Izuho Hatada[2], Yoshio Fujitani [1]*

1 Laboratory of Developmental Biology and Metabolism, Institute for Molecular and Cellular Regulation (IMCR), Gunma University, Maebashi, Japan, 2 Laboratory of Genome Science, Biosignal Genome Resource Center, Institute for Molecular and Cellular Regulation, Gunma University, Maebashi, Japan, 3 Metabolic Signal Research Center, Institute for Molecular and Cellular Regulation, Gunma University, Maebashi, Japan, 4 Department of Endocrinology, Research Institute of Environmental Medicine, Nagoya University, Nagoya, Japan, 5 Laboratory of Metabolic Regulation and Genetics, Institute for Molecular and Cellular Regulation, Gunma University, Maebashi, Japan

* fujitani@gunma-u.ac.jp

## Abstract

Understanding the mechanisms that regulate cell identity acquisition and cell proliferation is crucial, not only for elucidating cellular functions but also for clarifying the pathogenesis of various diseases, including neoplasms. Pancreatic endocrine cells typically express a single hormone, and their numbers are tightly regulated. Contrary to this general principle, in this study, we found that proglucagon-deficient mice exhibit a significant increase in the number of glucagon (GCG) and pancreatic polypeptide (PP) double-positive cells (GCG+ PP+ cells), together with the hyperplasia of both PP and α cells. Similarly, glucagon receptor-deficient mice displayed PP-cell hyperplasia and an increased number of GCG+ PP+ cells, with PP-cell replication implicated in this process. mTOR signaling was activated in GCG+ PP+ cells, suggesting its involvement in endocrine differentiation. Furthermore, impaired hepatic GCG signaling led to elevated plasma amino acid levels, which in turn promoted pancreatic endocrine cell proliferation and disrupted the maintenance of cellular identity in mice. Moreover, we found that increased glutamine levels promote GCG+ PP+ cell formation via mTOR signaling, revealing a novel regulatory mechanism underlying pancreatic endocrine cell plasticity. These findings provide new insights into endocrine cell regulation, and may contribute to a better understanding of pancreatic neuroendocrine tumor development, as well as the identification of novel therapeutic strategies.

**Data availability statement:** All relevant data are within the manuscript and its Supporting information files.

**Funding:** This work was supported by Grants-in-Aid for Scientific Research (JSPS KAKENHI; 18K08501 and 21K08547 to YN) from the Ministry of Education, Culture, Sports, Science, and Technology, Japan, and by the Takeda Science Foundation (2022088468 to YF). This research was also supported by the Platform Project for Supporting Drug Discovery and Life Science Research [BINDS] from AMED under Grant number JP21am0101120 (Support No. 95). The funders had no role in study design, data collection and analysis, decision to publish, or preparation of the manuscript.

**Competing interests:** The authors declare that they have no conflicts of interest associated with this study.

## Introduction

Each organ in the body precisely regulates both the number and types of its cells, to ensure proper function. Similarly, the pancreatic islets, which are the endocrine component of the pancreas, maintain a delicate balance among four types of cells, namely, α cells, β cells, δ cells, and pancreatic polypeptide (PP) cells. These cells secrete glucagon (GCG), insulin (INS), somatostatin (SST), and PP, respectively, working synergistically to regulate nutrient homeostasis, including that of glucose [1,2]. Disruptions in the regulation of cell proliferation or identity within the islets are closely linked to the onset of various diseases. For example, β-cell dysfunction or loss is a characteristic feature of type 1 and type 2 diabetes, resulting in insufficient INS action [3,4]. Conversely, the excessive proliferation of endocrine cells can result in endocrine neoplasms, which may impair normal pancreatic function and cause severe clinical complications [5,6]. Understanding the molecular mechanisms underlying cell fate determination and proliferation is therefore essential for maintaining the normal regulation of pancreatic islet function.

α cells, which are a type of endocrine cell, regulate their proliferation by secreting GCG, which modulates amino acid catabolism in the liver [7]. Consequently, mice lacking the proglucagon gene, which encodes GCG [8,9], or lacking the glucagon receptor [10–13] exhibit a common phenotype; i.e., enhanced α-cell proliferation and hyperplasia. Incidentally, the proglucagon gene encodes multiple proteins, including GCG, glucagon-like peptide-1 (GLP-1), and others. Although these proteins are derived from the same gene, they have distinct receptors. As a result, proglucagon gene-deficient mice lack all the encoded proteins, leading not only to GCG dysfunction but also to the dysfunction of GLP-1 and other proteins. Hyperplastic α cells caused by GCG dysfunction have been reported to develop into glucagonomas over time, and to metastasize to the liver [14–16]. The mechanism underlying α-cell hyperplasia in the context of GCG action deficiency is well characterized. Previous studies have shown that impaired GCG signaling in the liver induces α-cell hyperplasia [13,17–23]. This phenomenon is triggered by increased blood amino acid levels, particularly glutamine, owing to reduced amino acid uptake and catabolism in the liver. Increased glutamine levels upregulate the glutamine transporter Slc38a5 in α cells, promoting enhanced glutamine uptake and subsequent activation of the mTOR pathway, which is a key driver of α-cell hyperplasia [18–21]. Notably, treatment of hyperplastic α cells with the mTOR inhibitor rapamycin inhibits amino acid-dependent α-cell replication, and facilitates their transdifferentiation into β-like cells [18]. This suggests that GCG deficiency may enhance the plasticity of islet cells, enabling them to differentiate into other cell types. In addition to α cells, an increase in cells expressing the *Ppy* gene, which encodes PP, has been reported in GCG activity-deficient mice [8,23]. However, it remains unclear whether the number of PP protein-expressing cells actually increases, and how this is associated with the increase in α cells.

We previously reported our analysis of *Ppy*-lineage cells by lineage tracing experiments using *Ppy-Cre* mice [24]. We showed that approximately half of the α cells are derived from the *Ppy* lineage, suggesting that about half of the α cells express the

*Ppy* gene. Furthermore, we utilized *Ppy*-DTA mice, in which the diphtheria toxin A fragment is specifically expressed in *Ppy*-expressing cells, leading to their ablation. Interestingly, in the islets of these mice, not only were PP cells completely depleted, but approximately 80% of the α cells were also lost. These findings indicate that, under normal conditions, a substantial proportion of α cells express the *Ppy* gene, even if they do not produce the PP protein. This suggests that α cells share certain properties with PP cells. It is possible that the absence of PP protein expression in many α cells is regulated by a mechanism that suppresses PP protein translation, thereby contributing to the maintenance of α-cell identity.

Endocrine cells generally follow the "one cell, one hormone" principle, but some cells are known to express multiple hormones. In particular, during the early stages of human fetal development (8–15 weeks of gestation), hormone double-positive cells, which co-express multiple hormones, are frequently observed [25]. However, as fetal development progresses to later stages, the number of these cells decreases, suggesting that they represent an intermediate stage in endocrine cell differentiation. It is also suggested that immature cells may temporarily express multiple hormones during differentiation. Furthermore, in the mature pituitary glands of both fish and mammals, multi-hormonal cells, which produce and secrete multiple hormones simultaneously, have been identified. In fish, these cells are thought to function to enable the coordinated regulation of different hormones, which is necessary for adapting to changes in the aquatic environment [26]. Additionally, in the pancreatic islets of mature mice, it has been reported that some *PPY*-expressing cells co-express the *INS*, *GCG*, or *SST* gene [27]. These cells exhibit a mixed phenotypic profile, combining characteristics of β cells, α cells, and δ cells, and their gene expression profiles overlap with those of these endocrine cell types. Notably, under diabetic stress conditions, PP cells have been suggested to possess the plasticity to acquire insulin-producing capacity. However, the physiological significance of these multi-hormonal cells remains largely unknown.

In this study, we demonstrate that the hyperplasia of PP cells and GCG and PP double-positive cells (GCG$^+$ PP$^+$ cells) occurs in proglucagon-deficient mice. We aimed to identify the proteins encoded by the proglucagon gene that are responsible for inducing PP-cell hyperplasia and the increase in GCG$^+$ PP$^+$ cells, as well as to elucidate the underlying mechanisms.

## Materials and methods

### Animal care

The protocol for the animal experiments conducted in this study was formally approved by the Animal Care and Ethics Committee of Gunma University (approval number: D21-040). Mice were euthanized humanely by cervical dislocation following anesthesia with isoflurane (Fujifilm, Osaka, Japan), which was administered via a rodent anesthesia inhalation system (NARCOBIT-E, Natsume Seisakusho Co., Ltd., Tokyo, Japan). All efforts were made to minimize suffering. Mice were housed in a specific-pathogen-free barrier facility, maintained under a 12-hour light/dark cycle, and provided with unrestricted access to standard rodent chow (Oriental Yeast, Tokyo, Japan) and water.

### Mice

B6D2F1 mice were purchased from CLEA Japan (Kawasaki, Japan), ICR mice were obtained from Charles River Japan (Yokohama, Japan), and C57BL/6JmsSlc mice were obtained from Japan SLC, Inc. (Shizuoka, Japan). Male mice were used for all the experiments.

### Generation of *glucagon receptor* gene (*Gcgr*)-floxed mice

*Gcgr*-floxed mice were generated by the sequential electroporation method that we reported previously [28]. Sequences of target crRNAs flanking exons 4–13 of the *Gcgr* gene, and corresponding donor single-stranded oligodeoxynucleotides including loxP are shown in Table 1. Fertilized eggs were isolated from superovulated B6D2F1 female mice, 21 h post administration of human chorionic gonadotropin (hCG). The first electroporation to insert the 5′ loxP into intron 3 was

**Table 1. Sequences of target crRNAs flanking exons 4 to 13 of the *Gcgr* gene, and corresponding donor single-stranded oligodeoxynucleotides (ssODNs), including loxP.**

| Locus | Target | ssODN |
|---|---|---|
| Left loxP | GcgrL4 (5′- ATGGCAAGAT-CACTGGCCAG −3′) | GcgrL4loxP (5'-AAATGTTGCTGGGCTGGAGAGGGAGGTAGTCATG GCATCTAGATGGCAAGATCACTGGCCAATATTATAACTTCGTAT AATGTATGCTATACGAAGTTATAGGGGATGCCCCTTGTGG CCCATATCAGGGAGGCAGCCCCTTTAGATGCAGGCTCCGT GT-3') |
| Right loxP | GcgrR6 (5′- GAATTCCCT-GAGGACCTAGG −3′) | GcgrR6loxP (5'-GGCCATTTGTCACCAGGAAAGGCAAGGACCACCTCAT ACACTGAATTCCCTGAGGACCTAATAACTTCGTATAGCATA CATTATACGAAGTTATCTCGAGGGGGGACCTGACATCCAAC ACATGCAACCTTAATAATGGGGATGGAGATATGGTCATACC-3') |

conducted at the 1-cell stage (24–26 h post hCG); and the second electroporation to insert the 3′ loxP into intron 13 was conducted at the 2-cell stage (42–44 h post hCG). The sequentially electroporated embryos were then transferred to the oviducts of pseudopregnant ICR females. The floxed alleles of the obtained mice were confirmed by sequencing analysis. For PCR-based confirmation, the primers used for each Flox insertion site were as follows: for the left Flox insertion site, the forward primer was 5′-CTTGAAGGAGAGGTGCCTTG-3′, and the reverse primer was 5′-GGCCAGCAGGA GTACTTGTC-3′, and a 268-bp band is detected. For the right Flox insertion site, the forward primer was 5′-TCTTTCT CCAGGGTCTGCTG-3′, and the reverse primer was 5′-CAAGTTTCACCTTGGGTTCA-3′, and a 250-bp band is detected.

## Generation of systemic *Gcgr*-knockout mice and liver-specific *Gcgr*-deficient mice

The cytomegalovirus early enhancer and chicken β-actin (CAG) promoter-Cre transgenic mice (*CAG-Cre*) [29,30] were crossed with *Gcgr*-floxed mice. To confirm the deletion of exon 4 to exon 13 in the *Gcgr* gene by Cre recombinase, PCR was conducted using the primers 5′-CTTGAAGGAGAGGTGCCTTG-3′ and 5′-CAAGTTTCACCTTGGGTTCA-3′. If the deletion is present, a 155-bp PCR product was detected. Liver-specific *Gcgr*-knockout mice were generated by crossing *Gcgr*-floxed mice with albumin-*Cre* mice [31].

## Immunohistochemistry

After anesthesia with isoflurane (Fujifilm, Osaka, Japan), mice were perfused with normal saline through the heart to remove blood cells from the blood vessels. Subsequently, they were perfusion-fixed with 4% paraformaldehyde (PFA), and their pancreata were harvested. The harvested pancreata were fixed in 4% PFA at 4°C overnight. Subsequently, the pancreata were transferred to 70% ethanol. The pancreas samples were embedded in paraffin, and the paraffin pancreas blocks were sectioned at a thickness of 5 μm, and subjected to immunostaining. Immunofluorescence staining was conducted following previously described methods [32]. The primary antibodies used were guinea pig anti-INS (Ready-to-use, IR002, Dako-Agilent, Santa Clara, CA, USA), rabbit anti-GCG (1:1,000, ab92517, Abcam, Cambridge, UK), rabbit anti-Ki67 (1:200, ab16667, Abcam), rabbit anti-somatostatin (1:1,000, T-4103, Peninsula Laboratories, San Mateo, CA, USA), rabbit anti-phosphorylated S6 ribosomal protein (pS6) (1:200, 4858, Cell Signaling Technology, *Danvers*, MA, USA), mouse anti-PP (1:1,000, 23-2D3, IBL, Gunma, Japan) [32], and chicken anti-GFP (1:1,000, ab13970, Abcam). Tissue sections were incubated with the primary antibody, followed by a secondary antibody (Alexa Fluor 488, ab150077, Abcam; A11001 and A11039, Thermo Fisher Scientific, Waltham, MA, USA; Alexa Fluor 594, A11012 and A11005, Abcam; Alexa Fluor 647, A31573 and A31571, Abcam, all used at 1:1,000) together with the nuclear dye 4′,6-diamidino-2-phenylindole (DAPI, Dojindo Molecular Technologies, Kumamoto, Japan). The sections were then visualized using an FV1000 confocal laser-scanning microscope (Olympus, Tokyo, Japan).

## Quantification of α-cell and PP-cell mass

The quantification of α-cell mass and PP-cell mass was performed as previously described [24,33]. After anesthesia with isoflurane, mice were perfused through the heart with normal saline followed by 4% PFA, as described in the "Immunohistochemistry" section. Pancreata were then harvested, and nonpancreatic tissues were removed. The pancreata were post-fixed in 4% PFA at 4°C overnight. To obtain longitudinal sections from the tail to the head of the pancreas, the pancreas was embedded in paraffin, and paraffin blocks were prepared. From the entire pancreas, 6 sections per mouse were collected at 200-μm intervals, and immunostained with an anti-GCG or anti-PP antibody. The immunostained samples were analyzed using a BX-9000 microscope and the BZII analysis application software (Keyence Corporation, Osaka, Japan). The relative cross-sectional areas of α cells or PP cells were determined by dividing the cross-sectional area occupied by α cells or PP cells by the total cross-sectional area of the tissue. The total α-cell mass or PP-cell mass in the pancreas was calculated by multiplying the relative cross-sectional area of each cell type by the weight of the pancreas. Pancreata from four animals in each group were analyzed to determine the α-cell or PP-cell mass.

## Islet culture

Following previously reported methods, islets were isolated from 10-week-old mice [34]. The islets were collected using a micropipette under a dissection microscope, and cultured overnight in Roswell Park Memorial Institute (RPMI)-1640 medium (11875093, Thermo Fisher Scientific) supplemented with 1% penicillin/streptomycin. Subsequently, 100 islets were placed on a 24-well plate coated with Matrigel (Corning Incorporated, Corning, NY, USA), and cultured for 4 days. Islets in the control group were cultured in RPMI-1640 medium without L-glutamine (32404, Thermo Fisher Scientific) supplemented with 500 μM L-glutamine. Islets in the glutamine group were cultured in the same medium supplemented with 3,250 μM L-glutamine. After the culture period, the islets were fixed with 4% PFA, followed by treatment with 15% and 30% sucrose solutions. Finally, frozen tissue blocks were prepared using Optimal Cutting Temperature compound, followed by cryosectioning [32]. Details of the immunostaining procedure and antibodies used are provided in the "Immunohistochemistry" section above. The sections were subsequently visualized using an FV1000 confocal laser scanning microscope (Olympus).

## Mouse plasma collection

Prior to blood collection, the mice were weighed, and blood glucose levels were measured using the Glutest Neo Alpha blood glucose meter (Sanwa Kagaku Kenkyusho Co. Ltd., Aichi, Japan). Blood was collected by cutting the tails of the mice, and using a microhematocrit heparinized capillary tube (22–362566, Thermo Fisher Scientific). The collected blood was then transferred to a 1.5-mL tube. Subsequently, the collected blood was centrifuged at 6,200 rpm for more than 5 minutes, and the supernatant was transferred to a new 1.5-mL tube, and frozen using liquid nitrogen.

## Metabolomics analysis

Widely targeted metabolomics analysis was performed as previously described [35,36]. Briefly, frozen plasma samples (the sample preparation method is described above in the "Mouse plasma collection" section) in 1.5-mL plastic tubes were homogenized in 300 μL of cold methanol using φ3-mm zirconia beads and a freeze crusher (TAITEC Corp., Saitama, Japan) at 41.6 Hz for 2 min. The homogenates were mixed with 200 μL of methanol, 200 μL of $H_2O$, and 200 μL of $CHCl_3$, and vortexed for 20 min at room temperature. The samples were then centrifuged at 15,000 rpm (20,000 $g$) for 15 min at 4°C. The supernatant was mixed with 350 μL of $H_2O$, vortexed for 10 min at room temperature, and centrifuged at 15,000 rpm for 15 min at 4°C. The aqueous phase was collected and dried down using a vacuum concentrator. The samples were redissolved in 2 mM ammonium bicarbonate (pH 8.0) containing 5% methanol, and analyzed by LC-MS/MS.

Chromatographic separations in an Acquity UPLC H-Class System (Waters, Milord, MA, USA) were carried out under reverse-phase conditions using an ACQUITY UPLC HSS T3 column (100 mm × 2.1 mm, 1.8-μm particles, Waters) and

under HILIC conditions using an ACQUITY UPLC BEH amide column (100 mm × 2.1 mm, 1.7-µm particles, Waters). Ionized compounds were detected using a Xevo TQD triple quadrupole mass spectrometer coupled with an electrospray ionization source (Waters). The peak area of a target metabolite was analyzed using MassLynx 4.1 software (Waters).

## Statistical analyses

Statistical analyses were performed using GraphPad Prism version 9.1.1 software (GraphPad Software, San Diego, CA, USA). Results are presented as the mean ± standard error of the mean (SEM). Differences between two groups were assessed using a two-tailed unpaired Student $t$-test, and comparisons among three or more groups were performed using one-way ANOVA followed by the Tukey test. A $p$-value of less than 0.05 was regarded as indicating a statistically significant difference between groups.

## Results

### Induced hyperplasia of PP cells and formation of multi-hormonal cells in proglucagon gene-deficient mice

Proglucagon gene-deficient mice ($Gcg^{gfp/gfp}$) lack the production of all hormones encoded by the proglucagon gene, including GCG and GLP-1, owing to the knock-in of the $gfp$ gene into the proglucagon gene locus [8]. $Gcg^{gfp/gfp}$ mice are known to demonstrate α-cell hyperplasia similar to other mice with GCG signaling deficiencies [37]. In $Gcg^{gfp/gfp}$ mice, GFP is expressed in cells that would normally produce GCG. Furthermore, it has been reported that the expression level of $Ppy$ mRNA is significantly elevated in the pancreatic islets of these mice [8,23]. Based on this, we performed fluorescence immunostaining using the PP antibody that we developed [32] to investigate PP protein dynamics. Fluorescence immunostaining was performed on pancreatic islet sections from 40-week-old $Gcg^{gfp/gfp}$ mice. In these islets, in addition to α-cell hyperplasia, PP-cell hyperplasia was observed (Fig 1A). Furthermore, many GFP⁺ PP⁺ cells were detected. Since the expression level of $SST$ mRNA was also elevated [8], although not as prominently as $Ppy$ mRNA, we conducted triple staining using antibodies for PP, GFP, and SST. In the pancreatic islets of $Gcg^{gfp/gfp}$ mice, PP⁺ GFP⁺ SST⁺ cells were identified (Fig 1B and S1 Fig). In contrast, no PP⁺ GCG⁺ SST⁺ cells were observed in the pancreatic islets of $Gcg^{+/+}$ mice when triple staining was performed using antibodies for PP, GCG, and SST. These findings suggest that the absence of proteins encoded by the proglucagon gene induces PP-cell hyperplasia and the formation of multi-hormonal cells. As the proglucagon gene encodes multiple hormones, including GCG, it was unclear as to exactly which hormone deficiency caused the induction of PP-cell hyperplasia and the production of multi-hormonal cells.

### Increased PP cells in *Gcgr*-deficient mice

To test whether GCG, one of the proteins encoded by the proglucagon gene, is involved in the above effects, *CAG-Cre* mice were crossed with *Gcgr*-floxed mice to produce *Gcgr* knockout ($Gcgr^{Δ/Δ}$) mice (Fig 2A). We next investigated whether PP-cell hyperplasia and the production of GCG⁺ PP⁺ cells occur in $Gcgr^{Δ/Δ}$ mice at 10 weeks of age. Compared with $Gcgr^{+/+}$ control mice, $Gcgr^{Δ/Δ}$ mice demonstrated substantial PP-cell hyperplasia, and the presence of GCG⁺ PP⁺ cells (Fig 2B). Quantitative analysis demonstrated a significant increase in both PP-cell mass and the ratio of GCG⁺ PP⁺ cells relative to GCG⁺ cells in $Gcgr^{Δ/Δ}$ mice compared with $Gcgr^{+/+}$ mice (Figs 2C and D). Furthermore, PP⁺ GCG⁺ SST⁺ cells were also detected in $Gcgr^{Δ/Δ}$ mice at this stage, consistent with the findings shown in Fig 1B (S2 Fig). Collectively, these findings indicate that GCG signaling deficiency induces PP-cell hyperplasia and induces the formation of multi-hormonal cells, including GCG⁺ PP⁺ cells.

### Induction of PP-cell replication and an increase in GCG⁺ PP⁺ cells owing to GCG dysfunction

In 10-week-old $Gcgr^{Δ/Δ}$ mice, an increase in PP-cell mass and an increase in GCG⁺ PP⁺ cells were observed. To assess how these changes progress at earlier stages, we examined $Gcgr^{Δ/Δ}$ mice at 4 weeks of age. The PP cell mass in

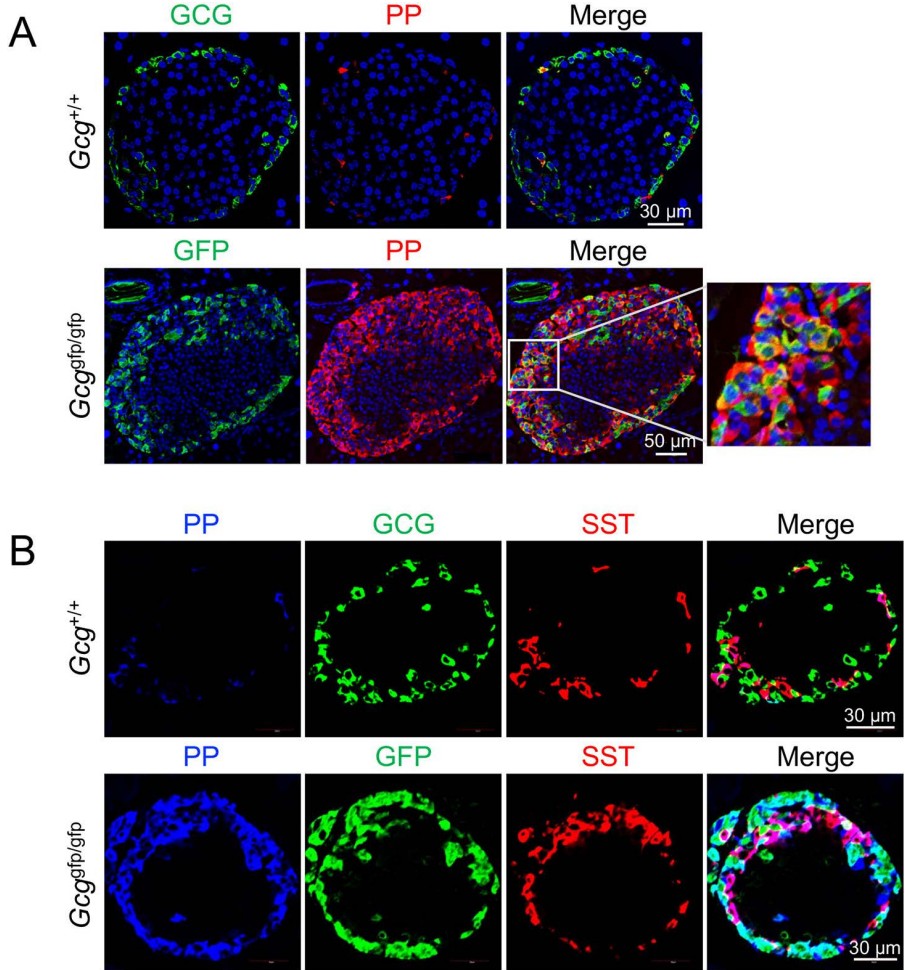

**Fig. 1. Deficiency of proteins encoded by the proglucagon gene induces PP-cell hyperplasia and the formation of multi-hormone-producing cells.** (A) Representative immunofluorescence staining of GCG or GFP (green), PP (red), and nuclei (DAPI; blue) in pancreas sections from 40-week-old $GCG^{+/+}$ mice and $Gcg^{gfp/gfp}$ mice. A magnified view of the boxed region is shown on the very right. (B) Representative immunofluorescence staining of PP (blue), GCG or GFP (green), and SST (red) in pancreas sections from 40-week-old $Gcg^{+/+}$ mice and $Gcg^{gfp/gfp}$ mice (B). Scale bars represent 30 µm or 50 µm.

4-week-old $Gcgr^{\Delta/\Delta}$ mice was significantly higher than that in $Gcgr^{+/+}$ mice (Fig 3A and B). However, unlike the comparison at 10 weeks of age (Fig 2B, right panel), the number of GCG⁺ PP⁺ cells in 4-week-old $Gcgr^{\Delta/\Delta}$ mice was not markedly increased compared with the control $Gcgr^{+/+}$ mice (Fig 3A, lower right panel). Therefore, we quantitatively compared the number of GCG⁺ PP⁺ cells in $Gcgr^{\Delta/\Delta}$ mice between 4 and 10 weeks of age. The data showed that at 4 weeks of age, the number of GCG⁺ PP⁺ cells was comparable to that in the control group. In contrast, at 10 weeks of age, the number of GCG⁺ PP⁺ cells was significantly higher in $Gcgr^{\Delta/\Delta}$ mice than in the control group (Fig 3C and D). Additionally, the number of GCG⁺ PP⁺ cells in 4-week-old $Gcgr^{\Delta/\Delta}$ mice was significantly lower than that in 10-week-old $Gcgr^{\Delta/\Delta}$ mice. However, since an increase in PP cell mass was observed at 4 weeks of age, the increase in PP cell mass was considered to be owing to factors other than an increase in double-positive cells. To investigate this possibility, we analyzed PP-cell proliferation. Costaining analysis was performed using antibodies against the proliferation marker Ki67, PP, and GCG to investigate whether PP-cell proliferation was induced. The results showed that the number of Ki67⁺ PP⁺ GCG⁻ cells was significantly

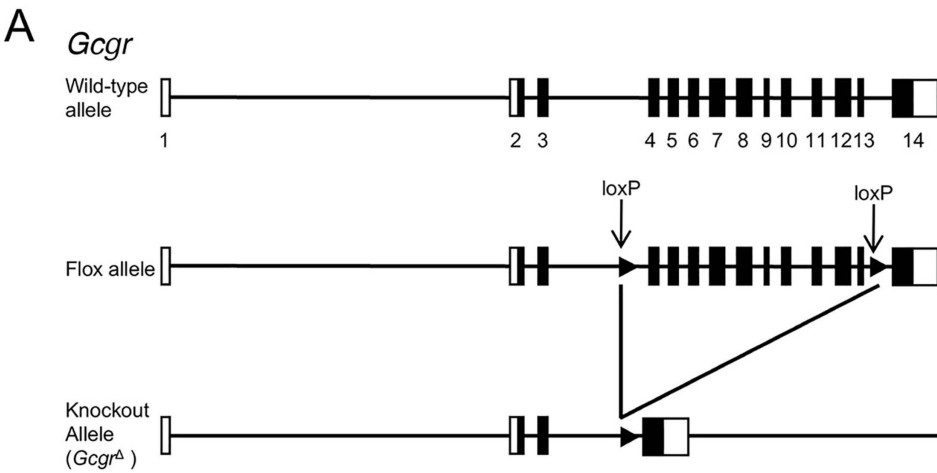

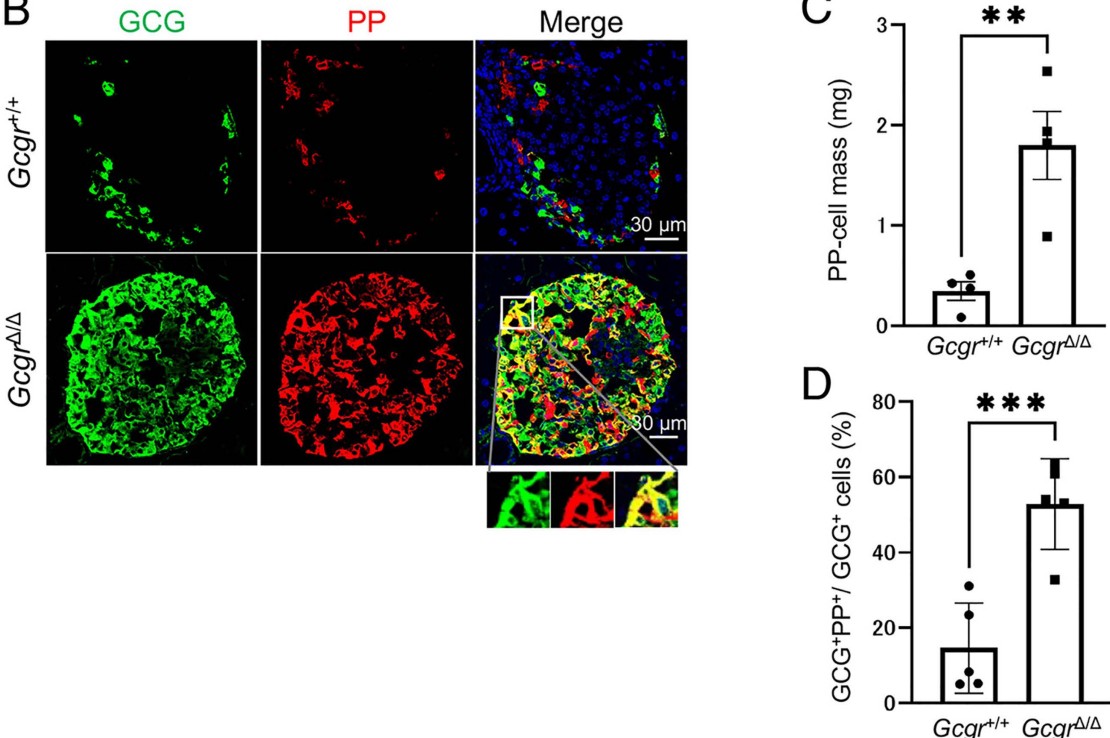

**Fig 2. GCG signaling deficiency induces PP-cell hyperplasia and the appearance of GCG+ PP+ cells.** (A) Schematic diagram of the loxP target site. (B) Representative immunofluorescence staining of GCG (green), PP (red), and nuclei (DAPI; blue) in pancreas sections from 10-week-old *Gcgr*⁺/⁺ mice and *Gcgr*Δ/Δ mice. Magnified images of the boxed region are shown at the bottom. Scale bar represents 30 μm. PP-cell mass (C) and the proportion of GCG⁺ PP⁺ cells among the GCG⁺ cells (D) in 10-week-old *Gcgr*⁺/⁺ mice and *Gcgr*Δ/Δ mice (n = 4–7 mice). Data are shown as the mean ± SEM, and was analyzed by the two-tailed unpaired Student *t*-test. **p < 0.01, ***p < 0.001.

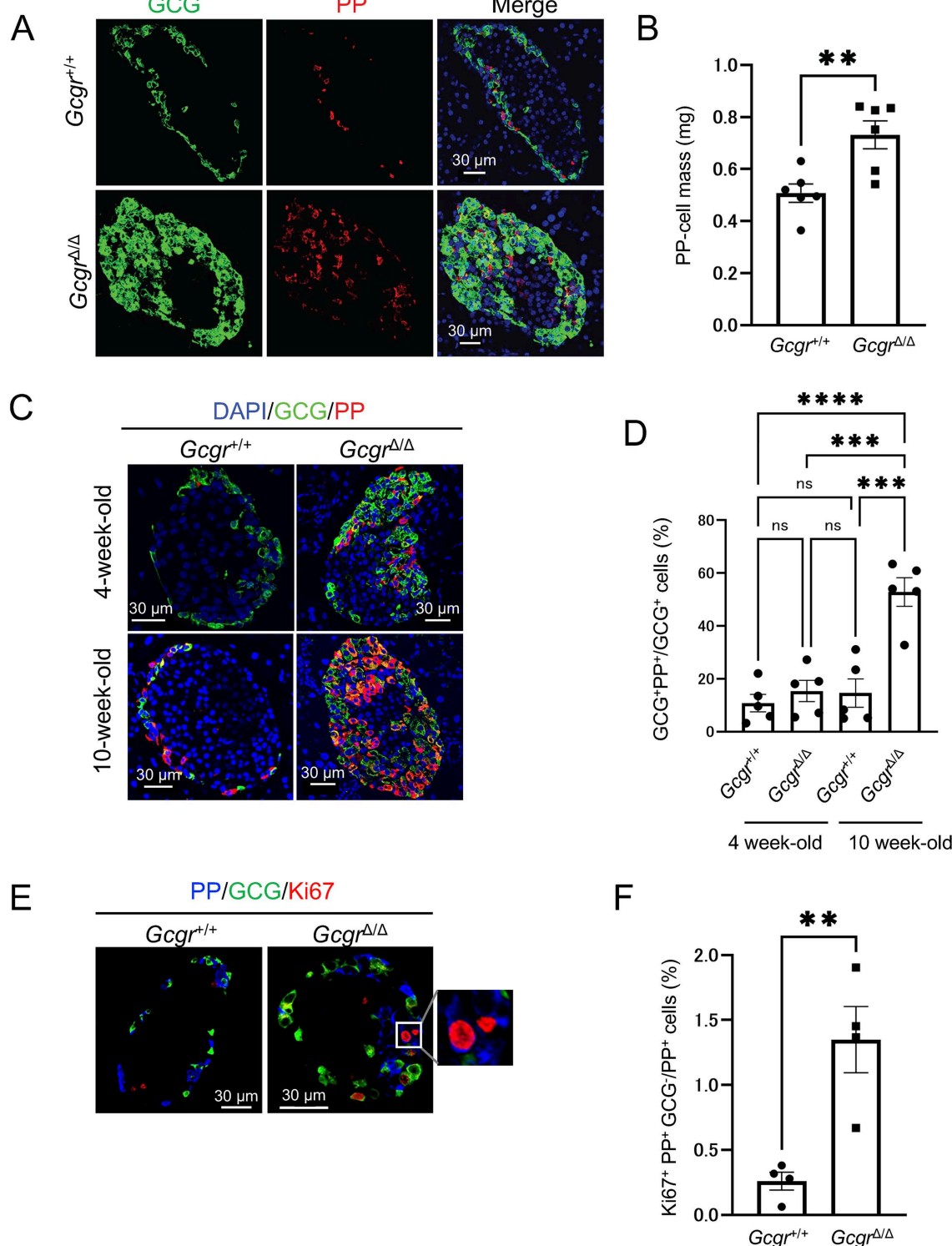

**Fig 3. PP-cell hyperplasia involves both the proliferation of PP cells and an increase in GCG+ PP+ cells.** (A) Representative immunofluorescence staining of GCG (green), PP (red), and nuclei (DAPI; blue) in pancreas sections from 4-week-old *Gcgr+/+* mice and *Gcgr^Δ/Δ* mice. Scale bar represents 30 μm. (B) PP-cell mass in 4-week-old *Gcgr+/+* mice and *Gcgr^Δ/Δ* mice (n = 6 mice each). (C) Representative immunofluorescence staining of GCG (green), PP (red), and nuclei (DAPI; blue) in pancreas sections from 4-week-old and 10-week-old *Gcgr+/+* mice and *Gcgr^Δ/Δ* mice. Scale bar represents 30 μm. (D)

Proportions of GCG$^+$ PP$^+$ cells among GCG$^+$ cells in 4-week-old and 10-week-old *Gcgr*$^{+/+}$ mice and *Gcgr*$^{\Delta/\Delta}$ mice (n = 4 mice each). (E) Representative immunofluorescence staining of PP (blue), GCG (green), and Ki67 (red) in pancreas sections from 4-week-old *Gcgr*$^{+/+}$ mice and *Gcgr*$^{\Delta/\Delta}$ mice. Scale bar represents 30 μm. A magnified image of the boxed region is shown on the very right. (F) Proportions of Ki67$^+$ PP$^+$ GCG$^-$ cells among PP$^+$ cells in 10-week-old *Gcgr*$^{+/+}$ mice and *Gcgr*$^{\Delta/\Delta}$ mice (n = 4 mice each). Data are shown as the mean ± SEM, and analyzed by the two-tailed unpaired Student *t*-test (B and F) or one-way ANOVA followed by the Tukey test (D). **$p < 0.01$, ***$p < 0.001$, ****$p < 0.0001$.

higher in *Gcgr*$^{\Delta/\Delta}$ mice than in *Gcgr*$^{+/+}$ mice (Figs 3E and F). These findings suggest that PP-cell proliferation is enhanced in *Gcgr*$^{\Delta/\Delta}$ mice. Furthermore, considering the temporal changes in PP-cell mass and the increase in GCG$^+$ PP$^+$ cells, it is likely that at 4 weeks of age, the increase in PP-cell mass is primarily owing to the enhanced replication of PP cells caused by GCG dysfunction, and that the increase in GCG$^+$ PP$^+$ cells subsequently becomes apparent at 10 weeks of age.

### Induction of PP-cell hyperplasia and an increase in GCG$^+$ PP$^+$ cells is caused by GCG dysfunction in the liver

Next, the tissue responsible for the hyperplasia of PP cells and the increase in GCG$^+$ PP$^+$ cells resulting from GCG dysfunction was investigated. Albumin-*Cre;Gcgr*$^{f/f}$ (Alb-*Cre;Gcgr*$^{f/f}$) mice, lacking GCG signaling specifically in the liver, which is the primary site of GCG action, were generated. As previously reported [13], 10-week-old Alb-*Cre;Gcgr*$^{f/f}$ mice demonstrated increased α-cell mass (S3 Fig). In addition, an increase in PP-cell mass and the presence of GCG$^+$ PP$^+$ cells were observed (Figs 4A–C). Further investigation of 40-week-old Alb-*Cre;Gcgr*$^{f/f}$ mice demonstrated a more pronounced increase in PP-cell mass (S4 Fig), as well as the presence of both GCG$^+$ PP$^+$ cells (Fig 4D). Previous studies have reported that mice with liver-specific GCG dysfunction exhibit α-cell hyperplasia. However, regarding β cells, while pancreatic enlargement leads to an increase in total β-cell mass, β-cell mass per gram of pancreas and INS secretion levels remain unchanged compared with control mice [13]. In this study, PP and INS double staining was performed using Alb-*Cre;Gcgr*$^{f/f}$ mice, revealing the presence of PP$^+$ INS$^+$ cells, which were not observed in control mice (Fig 4E). These results suggest that hepatic GCG dysfunction drives PP-cell hyperplasia and the emergence of GCG$^+$ PP$^+$ and INS$^+$ PP$^+$ cells.

### Induction of increased PP cell numbers and GCG$^+$ PP$^+$ cells owing to hyperaminoacidemia

Since impaired GCG action in the liver has been reported to elevate blood amino acid levels [18,20,21], we examined whether amino acid levels were altered in the *Gcgr*$^{\Delta/\Delta}$ and Alb-*Cre;Gcgr*$^{f/f}$ mice used in this study by measuring plasma amino acid concentrations. As expected, plasma amino acid levels, including glutamine, were significantly increased in 10-week-old *Gcgr*$^{\Delta/\Delta}$ and Alb-*Cre;Gcgr*$^{f/f}$ mice compared with control mice (S5 and S6 Figs). Next, we investigated whether the addition of large amounts of glutamine would increase the number of PP cells in the pancreatic islets and promote the formation of GCG$^+$ PP$^+$ cells (Fig 5A). When pancreatic islets were cultured with a high concentration of glutamine, the proportions of GCG$^+$ and PP$^+$ cells in the islets were significantly increased (Figs 5B and C). Moreover, the ratio of GCG$^+$ PP$^+$ cells among GCG$^+$ cells also increased (Fig 5D). To determine whether the increase in GCG$^+$PP$^+$ cells was mediated by the mTOR pathway, pancreatic islets were treated with the mTOR kinase inhibitor rapamycin. The glutamine-induced increase in GCG$^+$PP$^+$ cells was significantly suppressed by rapamycin treatment (Figs 5E and F). We then investigated whether pS6, a downstream target of mTOR signaling, could be detected in GCG$^+$PP$^+$ cells under glutamine-rich conditions. As shown in S7 Fig, pS6 signals were clearly observed in these cells. These findings indicate that elevated glutamine levels promote the expansion of GCG$^+$ cells, PP$^+$ cells, and GCG$^+$PP$^+$ cells in pancreatic islets through activation of the mTOR signaling pathway. Finally, we investigated whether mTOR was activated in GCG$^+$ PP$^+$ cells using tissues from *Gcgr*$^{\Delta/\Delta}$ mice. Consistent with the results from isolated islets, GCG$^+$ PP$^+$ pS6$^+$ cells were observed in these mice, supporting the role of mTOR activation in the formation of GCG$^+$ PP$^+$ cells (S8 Fig).

## Discussion

In this study, we analyzed liver-specific *Gcgr*-deficient mice, to gain additional insights into the effect of GCG deficiency, beyond the previously reported hyperplasia of α cells [10–13]. We found that impaired GCG signaling in the liver also

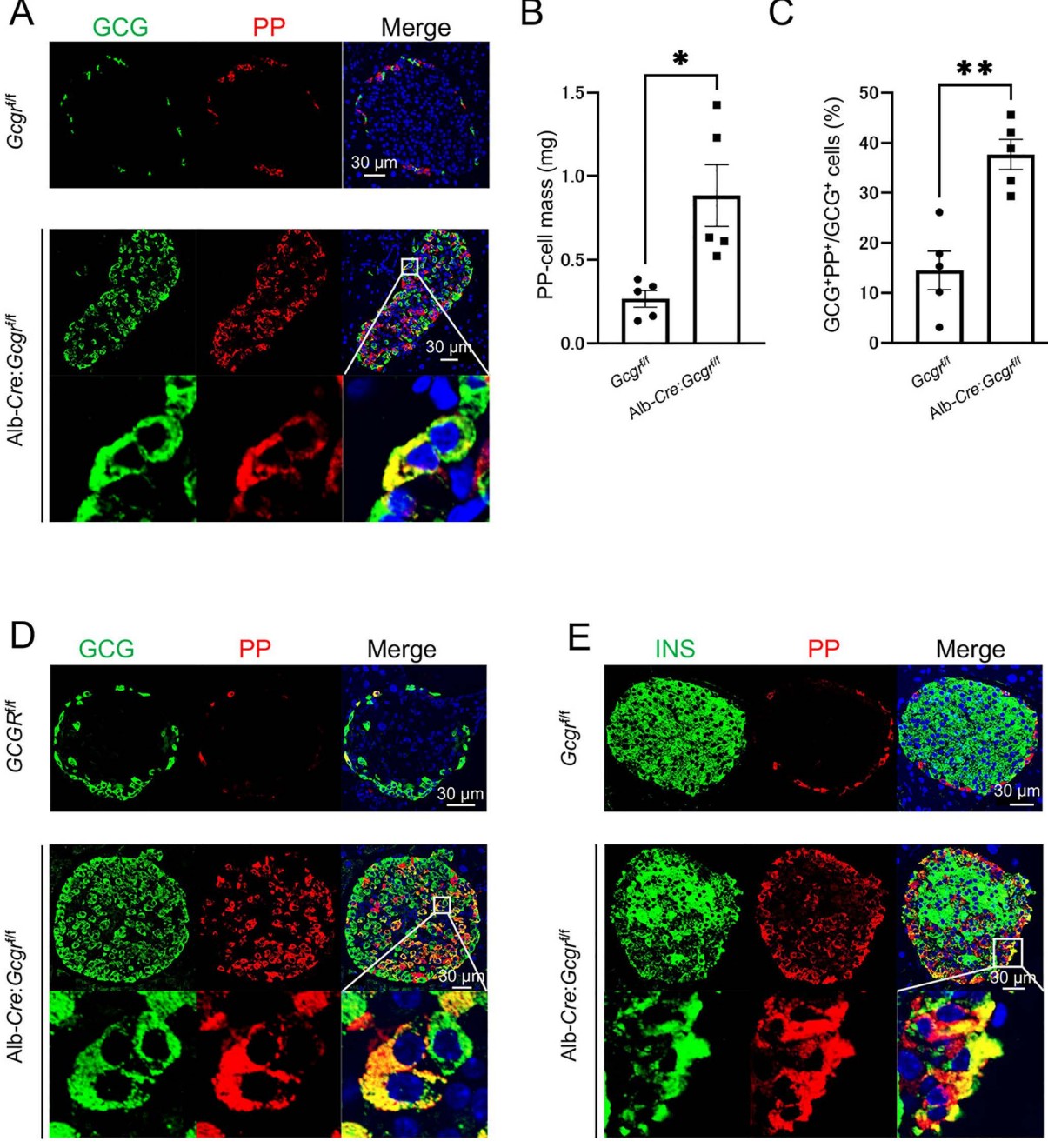

**Fig 4. GCG dysfunction in the liver induces PP-cell hyperplasia and the appearance of GCG+ PP+ cells.** (A) Representative immunofluorescence staining of GCG (green), PP (red), and nuclei (DAPI; blue) in pancreas sections from 10-week-old *Gcgr^f/f* mice and Alb-*Cre;Gcgr^f/f* mice. Scale bar represents 30 μm. The PP-cell mass (B) and the proportion of GCG⁺ PP⁺ cells among GCG⁺ cells (C) in 10-week-old *Gcgr^f/f* mice and Alb-*Cre;Gcgr^f/f* mice (n = 5 mice each) are shown. (D) Representative immunofluorescence staining of GCG (green), PP (red), and nuclei (DAPI; blue) in pancreas sections from 40-week-old *Gcgr^f/f* mice and Alb-*Cre;Gcgr^f/f* mice. (E) Representative immunofluorescence staining of INS (green), PP (red), and nuclei (DAPI; blue) in pancreas sections from 40-week-old *Gcgr^f/f* mice and Alb-*Cre;Gcgr^f/f* mice. Scale bar represents 30 μm. In (B) and (C), data are shown as the mean ± SEM, and analyzed by the two-tailed unpaired Student *t*-test. *$p < 0.05$, **$p < 0.01$. In C, D, and E, magnified images of the boxed regions are shown in the bottom row.

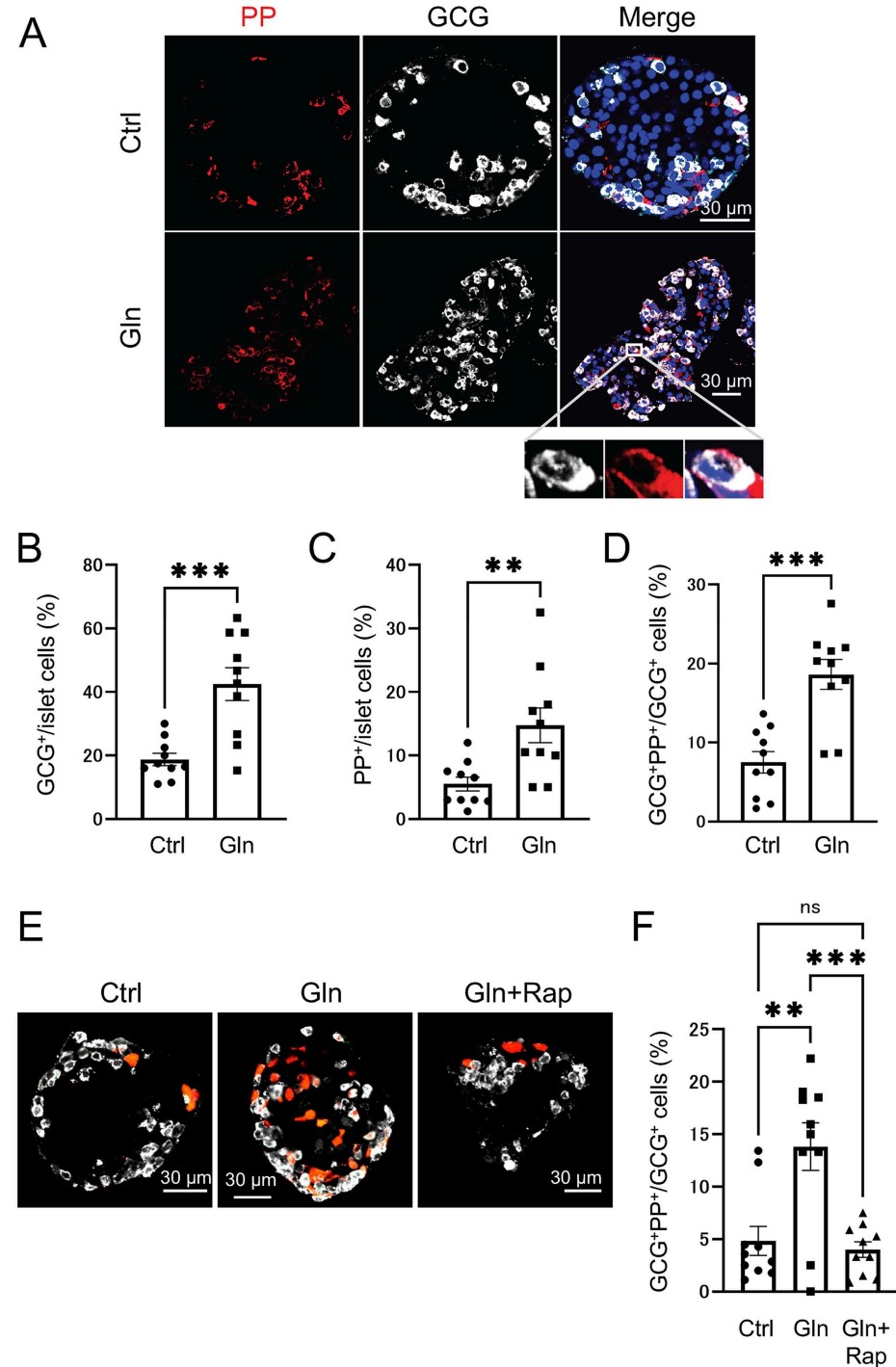

**Fig 5. GCG⁺ PP⁺ cells are increased in isolated islets upon incubation in high concentrations of glutamine, and this increase is mediated by the mTOR pathway.** (A) Representative immunofluorescence staining of PP (red), GCG (white), and nuclei (DAPI; blue) in pancreatic islets from 10-week-old wild-type mice, either untreated (Ctrl, upper panels) or incubated with a high concentration glutamine (Gln, lower panels). Scale bar represents 30 μm. Magnified images of the boxed region are shown at the bottom. The proportion of GCG⁺ cells among islet cells (B) (n = 10 islets), the proportion of PP cells among islet cells (C) (n = 10 islets), and the proportion of GCG⁺ PP⁺ cells among GCG⁺ islet cells (D) (n = 10 islets) from 10-week-old wild-type mice, either untreated (Ctrl) or incubated in a high concentration of glutamine (Gln). (E) Representative immunofluorescence staining of PP (red) and GCG (white) in pancreatic islets isolated from 10-week-old wild-type mice. Untreated islets (Ctrl, left panel), islets incubated with a high

concentration of glutamine (Gln, middle panel), and islets incubated with both a high concentration of glutamine and rapamycin (Gln+Rap, right panel) are shown. Scale bar represents 30 μm. (F) Proportions of GCG⁺ PP⁺ cells among GCG⁺ cells of the islets (n = 10 islets) from 10-week-old wild-type mice, which were either untreated (Ctrl), incubated with a high concentration of glutamine (Gln), or incubated with both a high concentration of glutamine and rapamycin (Gln+Rap). Data are shown as the mean ± SEM, and analyzed by the two-tailed unpaired Student *t*-test (B–D) or one-way ANOVA followed by the Tukey test (F). **$p < 0.01$, ***$p < 0.001$, ns, not significant.

leads to the hyperplasia of PP cells, and the emergence of GCG⁺ PP⁺ cells (Fig 4). Additionally, we observed a significant increase in plasma amino acid levels in these mice compared with the control groups (S5 Fig). Furthermore, by culturing islets isolated from wild-type mice with high concentrations of glutamine, one of the increased amino acids, we found that the increase in PP cells and GCG⁺ PP⁺ cells is facilitated by activation of the mTOR signaling pathway (Fig 5).

Previous studies have reported that expression of the *Ppy* gene is upregulated in proglucagon-deficient and *Gcgr*-deficient mice [8,23], however, whether PP protein expression is indeed increased in these mice remained unclear. In this study, as shown in Fig 2B, we observed a significant increase in PP protein expression owing to GCG signaling insufficiency, accompanied by a marked increase in PP-cell mass (Fig 2C). Smith et al. reported similar findings in their single-cell RNA sequencing study of islets from systemic *Gcgr*-deficient mice, in which they found a significant increase in PP-cell clusters in addition to α-cell clusters [23]. These results support the findings of our present study. Regarding the mechanism underlying the increase in PP cells, as illustrated in Figs 3E and 3F, systemic *Gcgr*-deficient mice demonstrated a higher proportion of Ki67-positive cells among PP⁺ cells that were not stained with GCG antibodies, compared with control wild-type mice. The number of PP cells was also higher in the *Gcgr*-deficient mice than in controls. These findings suggest that GCG signaling insufficiency promotes an increase in PP-cell mass through PP-cell proliferation. The mechanism underlying this PP-cell proliferation remains unclear, and represents a future research question. However, as shown in S9 Fig, in 4-week-old *Gcgr*ᐞ/ᐞ mice, pS6 signals were not detected in PP single-positive cells. This suggests that a different pathway, which is distinct from the mechanism by which GCG⁺ PP⁺ cells activate the mTOR signaling pathway to promote proliferation, may be involved. Previous studies have reported that other pancreatic endocrine cells, such as α and δ cells, also demonstrate enhanced proliferation under GCG signaling deficiency, although the mechanisms differ. Previous studies have shown that α cells increase in number through proliferation [37], whereas δ cells expand through both proliferation and duct-derived neogenesis [38]. In this study, we proposed that the increase in PP-cell numbers is driven by both proliferation and an increase in multi-hormone-producing cells. When comparing systemic *Gcgr*-deficient mice at 4 weeks of age (Fig 3B) with those at 10 weeks of age (Fig 2C), we observed a significant increase in PP-cell mass at 10 weeks of age. Corresponding to this substantial change, the number of GCG⁺ PP⁺ cells in 4-week-old *Gcgr*-deficient mice was comparable to that in wild-type mice (Fig 3D). However, in 10-week-old *Gcgr*-deficient mice, the number of GCG⁺ PP⁺ cells robustly increased compared with that in both wild-type mice and 4-week-old *Gcgr*-deficient mice (Fig 3D). We propose that this increase in GCG⁺ PP⁺ cells accounts for the substantial increase in PP-cell mass observed in 10-week-old *Gcgr*-deficient mice.

Owing to GCG dysfunction in the liver, the hyperplasia of PP cells and the emergence of multi-hormone-producing cells were observed (Fig 4), associated with an increase in blood amino acid levels. Similar to α-cell hyperplasia, the increased amino acids levels may have triggered these phenomena [18,20,21]. Supporting this hypothesis, culturing isolated islets in a high-glutamine medium resulted in an increased number of PP cells (Fig 5C) and an increase in GCG⁺ PP⁺ cells (Fig 5D). Furthermore, we compared PP-cell mass in 10-week-old systemic *Gcgr*-deficient mice (Fig 2D) and liver-specific *Gcgr*-deficient mice (Fig 4C). Despite being the same age, there was a significant difference in the PP-cell mass between the two groups. A potential explanation for this difference is the variation in plasma amino acid levels between the two groups. As shown in S5 Fig, plasma amino acid levels in systemic *Gcgr*-deficient mice were higher than those in liver-specific *Gcgr*-deficient mice. Particularly in terms of glutamine levels, systemic *Gcgr*-deficient mice demonstrated significantly higher values (S6 Fig). We believe that this difference in amino acid levels accounts for the variation in PP-cell mass of these mice at the same age.

Our results demonstrate that increased blood amino acid levels resulting from impaired GCG action in the liver activates the mTOR pathway in α cells, leading to the emergence of GCG+ PP+ cells. Previous studies have shown that forced expression of Aristaless-related homeobox (ARX) protein, a key transcription factor that determines the fate of α cells [39–41], in the embryonic pancreas or developing islet cells leads to a marked increase in α cells, accompanied by the loss of β and δ cells. Interestingly, under these conditions, the number of PP cells also increases substantially [39]. Humans with ARX mutations have normal numbers of β, δ, and ghrelin-secreting ε cells, whereas they are deficient in α and PP cells [42]. During development, α and PP cells differentiate from a common α/PP precursor [40,43,44]. This suggests that their differentiation may also depend on transcription factors other than ARX. However, the mechanisms that distinguish these two cell types and maintain their respective identities remain unclear. Since some transcription factors, such as ARX, are commonly active in both cell types, it is believed that under normal conditions, PP-protein expression is suppressed in α cells to preserve their identity. Collectively, our findings suggest that suppression of the mTOR pathway in α cells inhibits the expression of factors, including PP proteins, that drive PP-cell characteristics. Future studies should focus on elucidating how the mTOR pathway regulates these factors in α cells. Furthermore, we demonstrated that GCG dysfunction leads to the emergence of multi-hormone-producing cells, including PP+ INS+ cells (Fig 4E) and PP+ GFP+ SST+ cells (Fig 1B and S2 Fig). However, the precise mechanistic link between GCG dysfunction and the appearance of these cells remains unclear. Interestingly, a subset of pituitary cells is known to produce multiple hormones and function under specific physiological conditions, such as growth, the stress response, and reproduction. Gene expression analysis further suggests that these multi-hormone cells originate from undifferentiated or plastic cells [26]. Given these findings, one hypothesis is that the increase in blood amino acid levels due to impaired GCG action in the liver activates the mTOR pathway in α cells, leading to enhanced cell proliferation and increased cellular plasticity. Consequently, α cells may shift to a more undifferentiated state, resulting in an increased emergence of multi-hormone-producing cells, including GCG+ PP+ cells. Future studies are needed to elucidate how these multi-hormone-producing cells influence the progression and regulation of pathological conditions caused by GCG dysfunction.

One of the limitations of this study is the inability to determine the origin of GCG+PP+ cells. Although the Cre-loxP system employed here is widely used for lineage tracing, it has inherent limitations in labeling specificity. Off-target recombination can occur, which complicates the precise identification of cell origin. Consequently, we were unable to definitively trace the developmental lineage of GCG+PP+ cells using this approach. Another limitation is our inability to characterize the cellular properties of GCG+PP+ and PP+INS+ cells. To date, no specific surface markers have been identified that can reliably distinguish these double-positive cell populations. As a result, isolating them by cell sorting remains unfeasible. In future studies, identifying surface markers unique to each population will be essential for isolating these cells and comprehensively analyzing their functional characteristics. Such advances will be crucial for understanding the physiological relevance of these distinct endocrine cell populations.

In conclusion, impaired GCG signaling activates the mTOR pathway via hyperaminoacidemia, particularly elevated glutamine levels, inducing not only α-cell hyperplasia but also PP-cell hyperplasia and the formation of multi-hormone-producing cells (S10 Fig). Inhibition of mTOR suppresses these changes. This study reveals a novel mechanism by which GCG dysfunction alters pancreatic endocrine cell composition, highlighting the crucial role of mTOR in endocrine cell plasticity.

## Supporting information

**S1 Fig. Proportion of GCG+SST+ PP+ cells among GCG+ cells in 40-week-old *Gcg+/+* mice and *Gcggfp/gfp* mice.** Data are shown as the mean ± SEM, and analyzed by the two-tailed unpaired Student *t*-test. *$p < 0.05$; n = 4 each. (TIF)

**S2 Fig. Triple immunostaining using anti-GCG, anti-PP, and anti-SST antibodies in pancreatic islets of 10-week-old *Gcgr*$^{+/+}$ mice and *Gcgr*Δ/Δ mice.** (A) Representative immunofluorescence staining of GCG (green), SST (red), PP (white), and nuclei (DAPI; blue) in pancreatic sections from 10-week-old *Gcgr*$^{+/+}$ mice and *Gcgr*$^{Δ/Δ}$ mice. Magnified images of the boxed region are shown at the bottom. Scale bar represents 30 μm. (B) Quantification of the proportion of GCG$^+$SST$^+$PP$^+$cells among the GCG$^+$ cells in 10-week-old *Gcgr*$^{+/+}$ mice and *Gcgr*$^{Δ/Δ}$ mice (n = 4 each). Data are shown as the mean ± SEM, and analyzed by the two-tailed unpaired Student *t*-test. ***$p < 0.001$.
(TIF)

**S3 Fig. α-cell mass in 10-week-old *Gcgr*$^{f/f}$ mice and Alb-*Cre*:*Gcgr*$^{f/f}$ mice.** Data are shown as the mean ± SEM, and analyzed by the two-tailed unpaired Student *t*-test. *$p < 0.05$; n = 4 each
(TIF)

**S4 Fig. PP-cell mass in 40-week-old *Gcgr*$^{f/f}$ mice and Alb-*Cre*:*Gcgr*$^{f/f}$ mice.** Data are shown as the mean ± SEM, and analyzed by the two-tailed unpaired Student *t*-test. *$p < 0.05$; n = 6 each.
(TIF)

**S5 Fig. Plasma levels of individual amino acids in 10-week-old *Gcgr*$^{f/f}$, Alb-*Cre*:*Gcgr*$^{f/f}$, *Gcgr*$^{+/+}$, and *Gcgr*$^{Δ/Δ}$ mice.** The relative amounts shown on the Y-axis represent normalized values, where the mean *Gcgr*$^{f/f}$ level for each amino acid is set to 1. Therefore, it is not appropriate to directly compare these values between different amino acids. Data are shown as the mean ± SEM (n = 4–7 mice each).
(TIF)

**S6 Fig. Plasma glutamine levels in 10-week-old *Gcgr*$^{f/f}$, Alb-*Cre*:*Gcgr*$^{f/f}$, *Gcgr*$^{+/+}$, and *Gcgr*$^{Δ/Δ}$ mice.** The relative amounts shown on the Y-axis represent normalized values, where the mean *Gcgr*$^{f/f}$ level is set to 1. Data are shown as the mean ± SEM, and analyzed by one-way ANOVA followed by the Tukey test. *$p < 0.05$, **$p < 0.01$, ****$p < 0.0001$; ns, not significant.
(TIF)

**S7 Fig. Triple immunostaining of pancreatic islets from 10-week-old wild-type mice using anti-GCG, anti-PP, and anti-pS6 antibodies.** Representative immunofluorescence images of pancreatic islets isolated from 10-week-old wild-type mice stained for GCG (green), PP (red), and pS6 (white). Upper panels show untreated controls (Ctrl), and lower panels show islets incubated with high concentrations of glutamine (Gln). Scale bar: 30 μm. Enlarged views of the boxed regions are shown at the bottom.
(TIF)

**S8 Fig. Activation of the mTOR pathway in GCG$^-$PP$^+$ cells from 40-week-old *Gcgr*$^{+/+}$ mice and *Gcgr*$^{Δ/Δ}$ mice.** Representative immunofluorescence staining of GCG (green), pS6 (red), and PP (white) from 40-week-old *Gcgr*$^{+/+}$ mice and *Gcgr*$^{Δ/Δ}$ mice. Scale bar represents 30 μm. Enlarged images of the boxed region are shown on the very right.
(TIF)

**S9 Fig. Comparison of pS6 signal per PP single-positive cell in 4-week-old *Gcgr*$^{+/+}$ mice and *Gcgr*$^{Δ/Δ}$ mice.** (A) Representative immunofluorescence staining of GCG (green), pS6 (red), and PP (white) from 4-week-old *Gcgr*$^{+/+}$ mice and *Gcgr*$^{Δ/Δ}$ mice. Scale bar represents 30 μm. Enlarged images of the boxed region are shown on the very right. (B) Ratio of pS6$^+$ GCG$^-$ PP$^+$ cells per total PP$^+$ cells in 4 week-old *Gcgr*$^{+/+}$ mice and *Gcgr*$^{Δ/Δ}$ mice (n = 25 islets). Data are shown as the mean ± SEM, and analyzed by the two-tailed unpaired Student *t*-test. ns, not significant.
(TIF)

**S10 Fig. Graphical abstract of the study.** Impaired glucagon action in the liver leads to increased circulating amino acid levels, which in turn activate the mTOR signaling pathway in pancreatic α cells, promoting α-cell hyperplasia and an increase in GCG⁺PP⁺ cells. Simultaneously, this condition also induces the hyperplasia of pancreatic PP cells.
(TIF)

## Acknowledgments

We extend our heartfelt gratitude to W. Mizutani, Y. Tamura, A. Suda, N. Kagami, K. Iizuka, A. Fukaishi, Y. Miyazaki, H. Netsu, and T. Ishizaka of the Fujitani Laboratory (Gunma University, Gunma, Japan) for their invaluable technical assistance. We are also deeply grateful to Dr. H.A. Popiel (Tokyo Medical University, Tokyo, Japan) for critically reviewing the manuscript. Our sincere thanks go to the members of Dr. Y. Kitamura's laboratory and Dr. J. Shirakawa's laboratory (both from IMCR, Gunma University) for their insightful discussions. We also thank Dr. A. Oue, Dr. T. Kakinuma, and other members of the Bioresource Center at Gunma University for their generous support of general research on mice, and the Biosignal Genome Resource Center at Gunma University for the generation of *Gcgr*-floxed mice.

## Author contributions

**Conceptualization:** Yuko Nakagawa, Ayako Fukunaka, Takashi Sato, Yoshio Fujitani.

**Data curation:** Yuko Nakagawa.

**Formal analysis:** Yuko Nakagawa.

**Funding acquisition:** Yuko Nakagawa, Yoshio Fujitani.

**Investigation:** Yuko Nakagawa, Takuro Horii, Zhehao Zhang, Takashi Nishimura.

**Methodology:** Yuko Nakagawa, Yoshio Fujitani.

**Project administration:** Yoshio Fujitani.

**Resources:** Takuro Horii, Masaki Kobayashi, Tadahiro Kitamura, Yoshitaka Hayashi, Izuho Hatada.

**Supervision:** Yoshio Fujitani.

**Validation:** Yuko Nakagawa.

**Visualization:** Yuko Nakagawa.

**Writing – original draft:** Yuko Nakagawa, Yoshio Fujitani.

**Writing – review & editing:** Yuko Nakagawa, Yoshio Fujitani.

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
