## [Decision Letter · Decision Letter 0]

1 Apr 2025

PONE-D-25-12994Glucagon dysfunction in the liver induces hyperplasia of PP cells and the production of glucagon and pancreatic polypeptide double-positive cellsPLOS ONE

Dear Dr. Fujitani,

Thank you for submitting your manuscript to PLOS ONE. After careful consideration, we feel that it has great opportunity to publish in our Journal. However, we recommend some minor corrections to fully meet PLOS ONE’s publication criteria as it currently stands. Therefore, we invite you to submit a revised version of the manuscript that addresses the points raised during the review process.

We look forward to receiving your revised manuscript.

Kind regards,

Mohammad Sarif Mohiuddin, MD, PhD 

Academic Editor

PLOS ONE

“This work was supported by Grants-in-Aid for Scientific Research (18K08501 and 21K08547 to YN) from the Ministry of Education, Culture, Sports, Science, and Technology, Japan.”

“This work was supported by Grants-in-Aid for Scientific Research (18K08501 and 21K08547 to YN) from the Ministry of Education, Culture, Sports, Science, and Technology, Japan. We extend our heartfelt gratitude to W. Mizutani, Y. Tamura, A. Suda, N. Kagami, K. Iizuka, A. Fukaishi, Y. Miyazaki, H. Netsu, and T. Ishizaka of the Fujitani Laboratory (Gunma University, Gunma, Japan) for their invaluable technical assistance. We are also deeply grateful to Dr. H.A. Popiel (Tokyo Medical University, Tokyo, Japan) for critically reviewing the manuscript. Our sincere thanks go to the members of Dr. Y. Kitamura’s laboratory and Dr. J. Shirakawa’s laboratory (both from IMCR, Gunma University) for their insightful discussions. We also thank Dr. A. Oue, Dr. T. Kakinuma, and other members of the Bioresource Center at Gunma University for their generous support of general research on mice.”

“This work was supported by Grants-in-Aid for Scientific Research (18K08501 and 21K08547 to YN) from the Ministry of Education, Culture, Sports, Science, and Technology, Japan.”

Reviewers' comments:

Reviewer's Responses to Questions

**Comments to the Author**

1. Is the manuscript technically sound, and do the data support the conclusions?

Reviewer #1: Yes

Reviewer #2: Yes

2. Has the statistical analysis been performed appropriately and rigorously? 

Reviewer #1: Yes

Reviewer #2: Yes

3. Have the authors made all data underlying the findings in their manuscript fully available?

Reviewer #1: Yes

Reviewer #2: Yes

4. Is the manuscript presented in an intelligible fashion and written in standard English?

Reviewer #1: Yes

Reviewer #2: Yes

5. Review Comments to the Author

Reviewer #1: This manuscript provides an excellent investigation into the physiological link between glucagon signaling and pancreatic polypeptide (PP) cell hyperplasia using two or more different mouse models with glucagon signaling deficiency. However, there are a few minor issues in the structure and presentation of the manuscript that need attention. I would recommend revising these to improve the clarity and flow of the paper. Additionally, it might be beneficial to add a few more data or analyses to further support the conclusions, particularly in regard to the molecular mechanisms underlying the observed phenotypes.

In the Methods section, under "Immunohistochemistry," the fixation method through perfusion is described. However, there is an inconsistency between this description and the handling of tissues in the "Quantification of α- and PP-cell mass" section. Please revise the manuscript to ensure consistency between these two sections.

In the "Islet culture" section, there is insufficient information about how islets were collected. A brief additional explanation would be helpful. Also, at the end of this section, the embedding with OCT compound is mentioned, but there is no further description about immunostaining. Please either add this explanation to this section or include it in the "Immunohistochemistry" section.

In the "Metabolomic analysis" methods, there is no explanation about which tissues' "frozen samples" were used, and the Results section does not mention the results obtained using this method. This should be clarified and included.

In Figure 2B, the ratio of Gcg/PP double-positive cells is quantified. It would be helpful to perform the same quantification in Figure 1B for consistency. Additionally, in Figure 1B, SST staining is shown; please include SST staining in Figure 2B as well and provide the ratios of single, double, and triple-positive cells for each marker.

In the lines after 305, the term "self-proliferation" is used multiple times, but since it has not been evaluated whether the proliferation is truly "self" or not, it would be more appropriate to simply use the term "proliferation."

In Figures 3D and 3F, the ratios are used for quantification. However, as with Figure 3B, showing mass or the ratio relative to the total islet cells would help enhance the reader's understanding.

In Figure 5F, including the Ctrl+Rap group would make the comparison with the control group clearer.

Reviewer #2: 1)The authors identify GCG+ PP+ and PP+ INS+ cells but do not assess whether these cells are functional. Do they secrete both hormones?

Authors can try hormone secretion assays like ELISA for Glucagon or PP after sorting the cells.

2)The origin of double-positive cells remains unclear. Are α cells adopting PP identity, or are they a distinct lineage?

3)While mTOR activation is implicated, the downstream effectors remain undefined. Authors should check at least mRNA expression of mTOR pathway targeted genes.

4)There is no information about the sex of the mice. Did the study design with both mice? If both types of mice are used for the study, then author should observe whether the phenotypes are sex dependent or not?

5)Provide a graphical model summarizing the proposed mechanism.

6)Although the current title is technically accurate, it is somewhat complex and may be difficult for readers to grasp easily. I recommend simplifying the title to enhance clarity and accessibility, particularly for a broad, interdisciplinary readership. Consider rephrasing to more directly convey the central findings, such as: “Liver-specific glucagon dysfunction promotes PP-cell hyperplasia and formation of glucagon+ PP+ endocrine cells.”

6. PLOS authors have the option to publish the peer review history of their article (what does this mean? ). If published, this will include your full peer review and any attached files.

**Do you want your identity to be public for this peer review?** For information about this choice, including consent withdrawal, please see our Privacy Policy .

Reviewer #1: No

Reviewer #2: **Yes: ** Rashu Barua

---

## [Editor Report · Decision Letter 1]

11 Jul 2025

Liver-specific glucagon dysfunction promotes PP-cell hyperplasia and formation of glucagon and PP double-positive cells

PONE-D-25-12994R1

Dear Dr. Fujitani,

We’re pleased to inform you that your manuscript has been judged scientifically suitable for publication and will be formally accepted for publication once it meets all outstanding technical requirements.

Kind regards,

Mohammad Sarif Mohiuddin, MD, PhD 

Academic Editor

PLOS ONE
---

## [Editor Report · Acceptance letter]

PONE-D-25-12994R1

PLOS ONE

Dear Dr. Fujitani,

I'm pleased to inform you that your manuscript has been deemed suitable for publication in PLOS ONE. Congratulations! Your manuscript is now being handed over to our production team.

Kind regards,

on behalf of

Dr. Mohammad Sarif Mohiuddin

Academic Editor

PLOS ONE